# Clinical and growth outcomes of severely malnourished children following hospital discharge in a South African setting

Angelika Grimbeek[1][☉]*, Haroon Saloojee[2][☉]

1 Department of Paediatrics and Child Health, University of the Witwatersrand, Johannesburg, South Africa, 2 Division of Community Paediatrics, Department of Paediatrics and Child Health, University of the Witwatersrand, Johannesburg, South Africa

☉ These authors contributed equally to this work.
* angepeczak@gmail.com

**Data Availability Statement:** All relevant data are within the paper.

**Funding:** The authors received no specific funding for this work.

## Abstract

### Background

Data on outcomes of children with severe acute malnutrition (SAM) following treatment are scarce with none described from any upper-middle-income country. This study established mortality, clinical outcomes and anthropometric recovery of children with SAM six months following hospital discharge.

### Methods

A prospective cohort study was conducted in children aged 3–59 months enrolled on discharge from two hospitals in the Tshwane district of South Africa between April 2019 and January 2020. The primary outcome was mortality at six months. Secondary outcomes included relapse rates, type(s) and frequency of morbidities experienced and the anthropometric changes in children with SAM following hospital discharge. Standard programmatic support included nutritional supplements.

### Results

Forty-three children were enrolled with 86% of participants followed up to six months. Only a third of the participants had normal anthropometry at hospital discharge–a quarter still had ongoing SAM. There were no deaths, although four children (9%) were re-hospitalised including two for complicated SAM. Mean weight-for-length z-scores (WLZ) and wasting rates improved at one month but deteriorated by three months. At three months, six children (14%) either had ongoing or relapsed SAM–a SAM incidence rate of 20 per 1000 person-months despite more than half of the participants still receiving nutritional supplements at the time. Risk factors associated with persistent malnutrition at three months included a low WLZ on admission (relative risk [RR] 3.3, 95% confidence interval [95%CI] 1.2–9.2), being discharged from hospital before meeting WHO SAM treatment discharge criteria (RR 5.3, 95%CI 1.3–14.8) or having any illness by three months (RR 8.6, 95%CI 1.3–55.7).

**Competing interests:** The authors have declared that no competing interests exist.

## Conclusion

Post-discharge mortality and morbidity was lower than in other less resourced settings. However, anthropometric recovery was poorer than expected. Modifying discharge criteria, optimising the use of available nutritional supplements and better integration with community-based health and social services may improve outcomes for children with SAM post-hospitalisation.

## Introduction

Most programmes managing severe acute malnutrition (SAM) in children focus on immediate recovery only, resulting in post-discharge outcomes, such as mortality, ongoing SAM, anthropometric regression and clinical relapse being poorly understood and underestimated [1]. Relapse rates post-discharge from treatment for wasting (SAM) has been identified as one of the top five global research priority questions to achieve scale-up of treatment of wasting [2].

Studies conducted in Ethiopia, Malawi, Nigeria, Zimbabwe, Zambia, Bangladesh, and India have described mortality rates ranging between 0.8–10.3% from discharge to up to one year after either hospital or community-based management of SAM [3–9]. Factors associated with death included younger age (<12 months) and greater malnutrition severity at admission [3–9]. Mortality was highest among children living with HIV [4–6].

A 2019 systematic review indicated that 0–37% of children treated for SAM relapsed with most cases occurring within six months of discharge from the treatment programme. Common risks for relapse included lower anthropometric scores at hospital admission and on discharge along with illness simultaneous to the time of relapse [1].

Malnutrition remains a significant problem in South Africa, despite the country's status as an upper-middle-income country. SAM was a direct or contributory cause of a third of deaths in children under 5 years in 2013 [10]. There has been a reduction in SAM in-patient mortality at South African hospitals from 19.3% in 2010 to 7.1% in 2019 [11]. However, there is no national, district or institutional data on outcomes following discharge.

The study primarily aimed to establish mortality rates prospectively in children with SAM following hospital discharge. Secondary outcomes included relapse rates, the frequency and type(s) of morbidities experienced, and anthropometric changes in these children. It further aimed to identify factors associated with poorer long-term growth, particularly those that might be amenable to change or that could assist in identifying children requiring more targeted support. Lastly, it explored the uptake and benefit of local SAM support services.

## Methods

### Study setting and design

This was a prospective cohort study of children aged 3–59 months with SAM following their discharge from a regional (Mamelodi) or district (Bronkhorstspruit) hospital in the Tshwane district of Gauteng, South Africa. The catchment area of these two hospitals consists of urban and peri-urban sub-districts, including various townships. Enrolment occurred from April 2019 to January 2020, with children followed up for six months post-discharge.

### Study enrolment

All participants were hospitalised and had confirmed SAM on admission. SAM was defined as (a) a weight-for-length or -height z-score (WLZ) less than -3 on WHO growth standards, or (b) a mid-upper arm circumference (MUAC) of less than 11.5 cm, or (c) the presence of

bilateral pedal oedema. For children less than 6 months of age, a WLZ of less than -3 or the presence of bilateral pitting oedema was used for SAM diagnosis, with MUAC not considered. Moderate acute malnutrition (MAM) was defined as a WLZ between -2 and -3 or a MUAC between 11.5 cm and 12.5 cm. Treatment was based on the South African guidelines for in-patient SAM care [12].

Participants aged 3 to 60 months were identified by the researcher (AP) while being treated for SAM and children surviving to discharge were consecutively enrolled into the study. Study participants hospitalised more than once during the study period were only enrolled during the first admission. No exclusion criteria were applied. Children with chronic illnesses (such as HIV) or disabilities (such as cerebral palsy) were not excluded since these children are especially vulnerable to SAM and although non-nutrition related factors may contribute to their malnourishment, their outcomes post-discharge are as pertinent. The primary outcome was mortality at six months post discharge.

Children were discharged from in-patient care when they met the national discharge criteria—return of appetite, pedal oedema resolved, clinically well and alert, all medical complications resolved, and persistent and good weight gain for five consecutive days [12]. Discharge decisions were made by the treating clinicians.

Demographic, household and medical information was obtained through verbal primary caregiver interviews, while anthropometric (birth, admission and discharge) and clinical details were obtained from participants' case records and Road to Health booklets. If a child was born preterm (<37 completed weeks of pregnancy) their corrected age was charted throughout. A one-month first follow up date was set based on routine hospital practice.

On discharge, all participants received a nutrition supplement in the form of a commercial infant formula or paediatric milk powder, ready to use therapeutic food (RUTF), children's porridge or a combination of these, as per hospital protocol. Supplementation was intended to be continued until the patient's weight returned to normal and was maintained for three consecutive months. Supplementation could be continued for more than three months at the treating dietitian's discretion.

Applications were submitted for all South African nationals for participation in the South African Social Security Agency (SASSA) Zero Hunger project, a six-month programme enabling households with a malnourished child to receive monthly food parcels. All but one child was referred to the district ward based outreach team (WBOT) (community health care workers) for home visitations. The WBOT was expected to conduct at least one home visit and further follow up visits if necessary. Most children (60%) were referred to various social and health care specialists for added support. These included physiotherapists (67%), social workers (47%), occupational therapists (28%) and speech therapists (7%).

During the study period, monthly sub-district level meetings to reduce SAM mortality and improve the continuum of care through multi-disciplinary team work were ongoing. These were established in 2017 and discontinued in November 2019 when programme objectives of lower levels of SAM case fatality and appropriate communication networks were achieved. The meetings were led by the District Health Specialist Team and sub-district dietitians and aimed to audit SAM cases and better integrate SAM care by creating relationships between relevant stakeholders (clinic, hospital, SASSA and WBOT staff). However, there was poor engagement by SASSA representatives and WBOT members.

### Follow up procedure

Participants were expected to attend institutional follow-up sessions at one and three months post-discharge when anthropometry (weight, length and MUAC) was undertaken, and

information about any morbidities experienced and pertinent medical and social changes attained. The follow up occurred either at the facility in which the child had been treated or at a down-referral clinic.

Participants were evaluated by the researcher or a dietitian. If a caregiver failed to attend a scheduled visit, she was contacted telephonically and all information, except anthropometry, collected. A third and final follow up session was conducted telephonically six months after discharge.

Complete follow-up at one and three month visits was defined by the physical presence of the participant and the follow-up deemed partial if only a telephonic interview was possible. Participants unreachable after three attempts within three weeks of the scheduled visit date were classified as 'lost to follow-up' for that visit. All participants were contacted at six months regardless of whether they attended the one and three month follow-up visits.

At follow-up visits, weight was measured to the nearest 100 g, using an electronic scale, length/height to the nearest 0.5 cm using a measuring board and MUAC to the nearest 1 mm, using WHO/UNICEF colour-coded tapes. A child was considered to have no acute malnutrition (NAM) if they had a WZL $\geq$ -2 and MUAC $\geq$ 12.5 cm, or MAM or SAM as defined previously [12]. Participants were regarded as relapsed, when there was a recurrence of SAM (as defined at study onset) [2]. Persistent malnutrition was defined as the presence of MAM or SAM in a child who previously had SAM. Secondary growth measurements of underweight and stunting were measured using weight-for-age (WAZ) and height/length-for-age (HAZ).

## Sample size estimation

The study intended to enrol 50 participants, allowing for detection of 10% mortality with a 95% confidence interval of 3–22%. A 10% mortality rate at six months following discharge was predicted based on the 0.8–10.3% mortality range identified in previous studies [3–9]. The actual sample size was, however, limited by fewer than anticipated children with SAM being admitted during the study period.

## Data analysis

Data was entered into Excel 2010 (Microsoft, Seattle, USA), cleaned and exported to STATIS-TICA 13.5 (Tibco, California, USA) for analysis. World Health Organization Anthro software v3.2.2 (WHO, Geneva, Switzerland) was used to calculate the WLZ, WAZ and HAZ. Differences in proportions were compared by the chi-squared test or Fisher exact test when cell numbers were <5. Strength of association was determined by calculating relative risks (RR) and related 95% confidence intervals (95% CIs), using IBM SPSS Statistical version 27.0 (SPSS Inc, Chicago, USA). A p-value <0.05 was considered statistically significant.

## Ethical considerations

The study was approved by the University of Witwatersrand's Human Research Ethics Committee (clearance certificate number: M180812) and the National Health Research Database (GP_201812_016). Signed consent for participation was obtained from the primary caregivers of participating children.

## Results

Forty-three children were enrolled, 32 at the regional and 11 at the district hospital. Follow up was 100% at one month (28% partial follow up), 89% at three months (33% partial follow up) and 86% of participants' caregivers were interviewed at six months.

## Baseline characteristics

Participants' baseline characteristics are shown in Table 1. Their median (IQR) age was 11 (8, 20) months with 11% aged between three and six months. More were female (63%). All were black African, and 86% were South African nationals. All but one of the participant's primary caregiver was their biological mother. Two-thirds came from a single parent household. Most caregivers had no fixed monthly income (77%) and were recipients of a government social grant (79%). Half the children lived in informal houses constructed of corrugated iron. Water sources varied with 42% of participants reporting an indoor tap, 21% an outside tap and 37% using a communal tank or tap. Two-thirds of homes had electricity and one-half had a fridge.

## Medical history

Forty percent of children had been previously admitted to hospital, most often for neonatal complications (65%), diarrhoea (12%) and pneumonia (6%). Only one child had a previous SAM admission. Three participants had a long-term disability (cerebral palsy). None had any contractures preventing adequate length measurements allowing use of WHO growth standards. There was high HIV exposure—19 of the caregivers (44%) were HIV positive, all of whom were on antiretroviral treatment. Five children (12%) were HIV positive and were initiated or continued on antiretroviral therapy during hospitalisation. Co-morbidities during the admission included diarrhoea (35%), pneumonia (16%) and tuberculosis (7%).

## Admission status

At admission, one-half of the participants (49%) presented with a WLZ below -3, 40% had a MUAC less than 11.5 cm and 35% had bilateral pedal oedema. Ten children (23%) presented with a low WLZ and MUAC and three (7%) presented with a low MUAC and pedal oedema. No participant had both a low WZL and pedal oedema, nor did any have all three SAM criteria.

**Table 1. Baseline characteristics of participants discharged from SAM in-patient treatment (n = 43).**

| Characteristic | Frequency | Percent |
|---|---|---|
| Female | 27 | 63 |
| **Caregiver nationality** | | |
| South African | 37 | 86 |
| Mozambican | 2 | 5 |
| Zimbabwean | 4 | 9 |
| **Primary caregiver** | | |
| Mother | 42 | 98 |
| Grandmother | 1 | 2 |
| **Secondary caregiver** | | |
| Father | 18 | 42 |
| Grandmother | 9 | 21 |
| Grandfather | 1 | 2 |
| Aunt | 1 | 2 |
| None | 14 | 33 |
| **Social support** | | |
| Child support grant | 31 | 72 |
| Care dependency grant | 1 | 2 |
| Zero Hunger Project | 1 | 2 |
| None | 10 | 23 |

## Discharge status

Since discharge was based on clinical criteria and not the extent of anthropometric recovery, only a third of the participants had normal anthropometry (no acute malnutrition [NAM]) at discharge. Forty-two percent of the children were discharged with MAM and one-quarter had ongoing SAM. Of the MAM group, 33% had a low WLZ, 39% a low MUAC and 28% had both. In the SAM group, 36% qualified based on WLZ alone, 9% on a low MUAC and over half (55%) had both.

## Mortality and morbidity

No participant died during the six-month follow up period, three children relapsed (recovered then regressed to SAM) for an incidence rate of 20 per 1000 person-months. Three children never recovered from SAM. Thus, six children (14%) either had ongoing or relapsed SAM. Two of these children were readmitted for complicated SAM and the other four managed as outpatients. One of the three relapsed children was less than six months old at enrolment and of the three ongoing SAM, two had cerebral palsy.

During the six months, over half (23/43 [53%]) of the participants were ill; (10/43, [23%]) had a serious acute illness (such as pneumonia, diarrhoea and tuberculosis). Four children were hospitalised, all for pneumonia (two had concurrent TB). Other common morbidities by three months included a cold or fever (13%) and diarrhoea (11%). One child was newly diagnosed with HIV and five were exposed to TB during the follow up period.

Of the three children with cerebral palsy, two had ongoing SAM until three months. One child was discharged as NAM and remained NAM at one month follow up with no anthropometry taken at three months (partial follow up). All three were still receiving nutritional supplements at six months.

Of the five children living with HIV, three were discharged with SAM. Two recovered to MAM by one month with no anthropometry taken at three months. One had ongoing SAM until three months. One child was discharged as MAM and recovered to NAM by one month, and one was discharged as NAM which continued until one month. None of the five children experienced any serious acute illness during the follow up period.

No statistically significant differences in relapse rates, anthropometric outcome at three months and illness patterns were noted when comparing children who had SAM but not HIV or CP, to children who had either of these conditions.

## Anthropometric recovery

Fig 1 depicts participants' anthropometric status at discharge and at one and three months follow-up. In 14 children with normal discharge anthropometry (NAM) there was one regression to MAM at one month, but this corrected by the three month follow up.

In the MAM at discharge group, 8/13 (62%) recovered to NAM by one month, with five regressing to MAM and two to SAM thereafter.

Of the six children discharged with SAM, three still had SAM at three months, two had MAM and one recovered. In summary, by three months, just over half the participants (54%) had NAM, 29% had MAM and 17% had ongoing SAM.

Fig 2 shows the WLZ of 24 individual children (with available data) at admission, discharge, and one and three months following discharge. In all participants, WLZ increased from admission to discharge. There was a continued increase in WLZ for most participants for the first month. However, almost half of the participants (46%) had a decrease in their WLZ between one and three months.

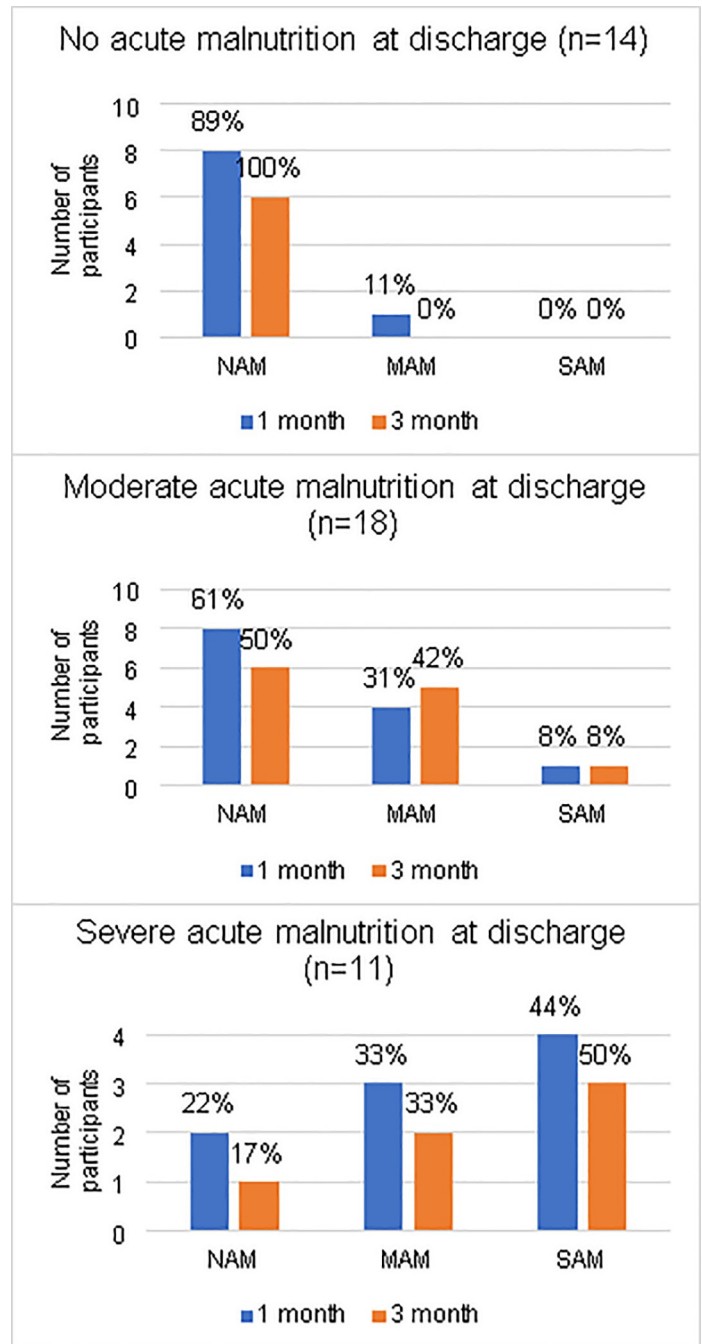

**Fig 1. Anthropometric status at discharge related to one and three month follow-up status.** Abbreviations: NAM = No acute malnutrition, MAM = Moderate acute malnutrition, SAM = Severe acute malnutrition.

Weight gain following discharge was calculated at one and three months. At one month, the median (IQR) weight gain since discharge was 2.9 (0.9, 4.3) g/kg/day, with a reduction to 1.6 (0.2, 2.6) g/kg/day between one to three months. The median weight gain was 1.4 (0.4, 2.0) g/kg/day over the three month period.

Table 2 shows the anthropometric status of children at the different follow up periods. Over three months, the WAZ, HAZ and MUAC improved. Stunting was present in 72% of the

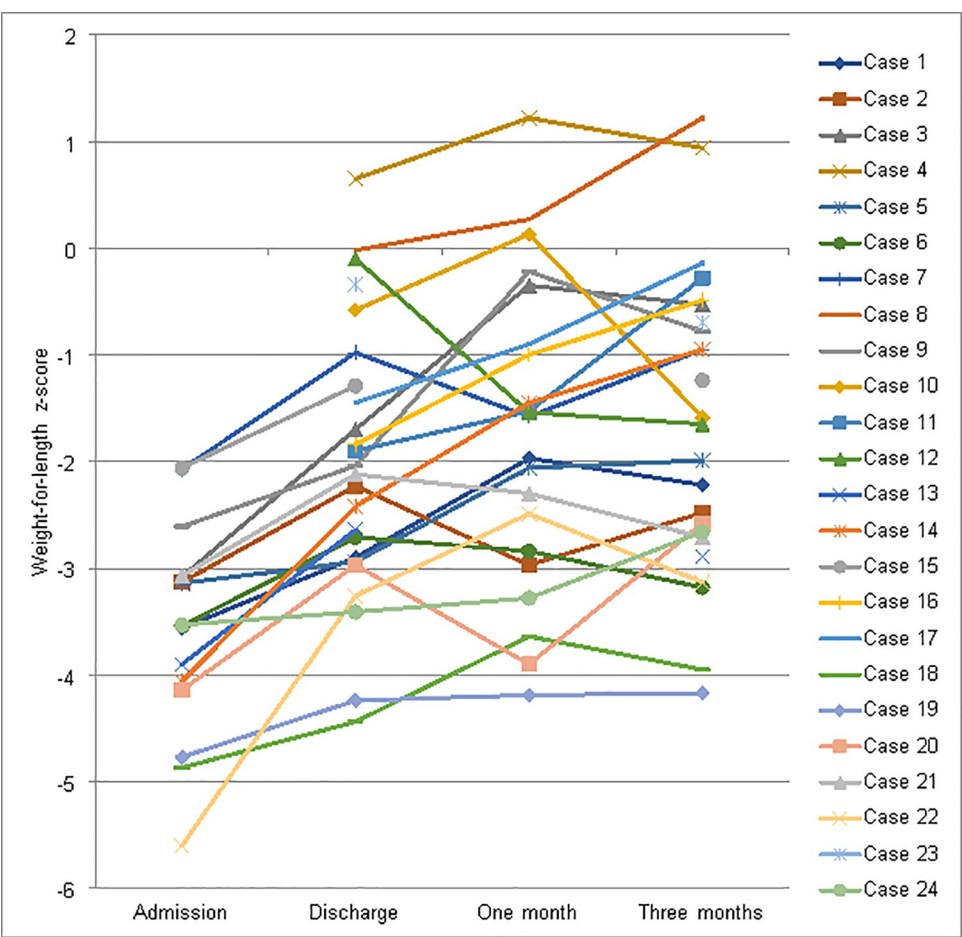

**Fig 2. Weight for length/height z-scores of individual children at admission, discharge, and one and three months following discharge (n = 24).**

participants at discharge with 42% severely stunted; 63% of the children were still stunted at three months. The mean WLZ and wasting rate improved at one month but deteriorated by three months. At discharge only 42% of participants had a normal MUAC (>12.5 cm) which improved to 77% and 79% at 1 and 3 months, respectively.

## Persistent malnutrition

Participants with persistent malnutrition (MAM or SAM) (n = 11, 46%) at three months were compared to those who fully recovered anthropometrically (n = 13, 54%). Persistent malnutrition at three months was significantly more likely to be present in children who had a WHZ <-3 compared to those that had a had a WHZ >-3 at hospital admission; (9/12 (75%) vs. 2/12 (17%), RR 3.3, (95% CI 1.2–9.2), p = 0.01) and less likely if admitted with pedal oedema (0/8 (0%) vs. 11/16 (69%), RR 0.31 (95%CI 0.15–0.65), p = 0.002). Participants discharged with ongoing SAM were trending to be more likely to remain malnourished (5/6 (83%) vs. 6/18 (33%), RR 4.0, (95%CI 0.65–24.7), p = 0.06) while those discharged as NAM were significantly less likely to be malnourished at three months (0/6 (0%) vs. 11/18 (61%), RR 0.36 (95%CI 0.22–0.69), p = 0.02).

Children who at the time of hospital discharge did not meet the WHO SAM treatment discharge criteria of either a MUAC of ≥12.5 or a WLZ of >-2 were more likely to be malnourished at three months (8/8 (100%) vs. 3/16 (19%), RR 5.3 (95%CI 1.9–14.8), p<0.001).

**Table 2. Anthropometry at discharge, one month and three month follow up.**

| | Admission n = varied* | | Discharge | | One month | | Three months n = 24 | |
| | | | n = 43 | | n = 31 | | | |
| | Mean (SD) | Proportion | Mean (SD) | Proportion | Mean (SD) | Proportion | Mean (SD) | Proportion |
| | | n (%) | | n (%) | | n (%) | | n (%) |
|---|---|---|---|---|---|---|---|---|
| Weight for age z-score (WAZ) | -3.97 (1.11) | | -2.95 (1.30) | | -2.46 (1.38) | | -2.38 (1.98) | |
| Moderately underweight | | 2 (7) | | 16 (37) | | 12 (39) | | 6 (25) |
| Severely underweight | | 25 (89) | | 18 (42) | | 9 (29) | | 8 (33) |
| Height/ length for age z-score (HAZ) | -2.53 (1.53) | | -2.68 (1.53) | | -2.66 (1.54) | | -2.28 (1.55) | |
| Moderately stunted | | 11 (26) | | 13 (30) | | 10 (32) | | 6(25) |
| Severely stunted | | 17 (40) | | 18 (42) | | 13 (42) | | 9 (38) |
| Weight for length z-score (WLZ) | -3.44 (0.98) | | -1.99 (1.20) | | -1.33 (1.47) | | -1.63 (1.42) | |
| Moderately wasted | | 5 (18) | | 13 (30) | | 6 (19) | | 6 (25) |
| Severely wasted | | 22 (79) | | 8 (19) | | 4 (13) | | 4 (17) |
| Mid-upper arm circumference (MUAC) [Median (IQR)] | 11.5 (11.0–12.9) | | 12.3 (11.8–12.8) | | 13.0 (12.5–14.5) | | 13.6 (12.6–14.2) | |
| Moderately wasted | | 6 (14) | | 16 (37) | | 5 (16) | | 5 (21) |
| Severely wasted | | 22 (51) | | 9 (21) | | 2 (7) | | 0 (0) |

*varied sample size. Admission WAZ & WLZ n = 28 due to fifteen participants with oedema. HAZ and MUAC n = 43.

Table 3 compared persistent malnutrition rates at three months compared to no acute malnutrition based on anthropometry at hospital discharge. The first criterion was based on participants discharged at different WLZ scores and did not consider the MUAC value. When participants were discharge with a WLZ of higher than -2.0 there was no persistent malnutrition at three months this increased to two children (15%) when the WLZ was greater than -2.2 at discharge. The second criteria included WLZ and MUAC of greater than 12.0 cm at discharge. There was no persistent malnutrition with a MUAC >12.0 cm and WLZ of either >-2.0 or >-2.1 however 1 child (11%) was persistently malnourished with a WLZ >-2.2. This indicates that the better the nutritional recovery at discharge the less likely there will be persistent malnourishment at three months.

A serious acute illness within three months was another factor associated with persistent malnutrition (4/4 (100%) vs 7/20 (35%), no RR estimate, p = 0.03) as was having any illness within three months (9/10 (90%) vs 2/14 (14%), RR 8.6 (95%CI 1.3–55.7), p = 0.001).

**Table 3. Persistent malnutrition rates at three months compared to no acute malnutrition based on anthropometry at hospital discharge.**

| Criterion | Persistent malnutrition | No acute malnutrition | p-value |
| | n (%) | n (%) | |
|---|---|---|---|
| **WLZ alone** | | | |
| >-2.0 | 0 (0) | 11 (100) | <0.001 |
| >-2.1 | 1 (8) | 11 (92) | 0.006 |
| >-2.2 | 2 (15) | 11 (85) | 0.003 |
| >-2.3 | 3 (21) | 11 (79) | 0.011 |
| **MUAC >12.0 cm and WLZ** | | | |
| >-2.0 | 0 (0) | 8 (100) | <0.001 |
| >-2.1 | 0 (0) | 8 (100) | <0.001 |
| >-2.2 | 1 (11) | 8 (89) | 0.013 |
| >-2.3 | 2 (22) | 7 (78) | 0.089 |

At one month, WBOT's visited seven children, all of whom were subsequently malnourished at three months, while of 17 children not visited only 4 (24%) were subsequently malnourished (p<0.001).

No statistical significant difference in persistent malnutrition rates was identified for any demographic or household factor. Neither the presence of any coexisting illness in the child or caregiver, nor other supportive care measures provided to the child proved to be a significant risk or benefit either. Participant HIV exposure or infection was also not related to persistent malnutrition.

## Nutritional supplements

Children were routinely offered either an age-appropriate commercial infant formula, porridge, ready-to-use therapeutic food (RUTF), or a combination of these. Selection of supplements was based on the availability of supplements at the hospitals or clinics and on the clinical judgement of the treating dietitian.

All children were provided with nutritional supplements at discharge. Almost three-quarters remained supplement recipients at one month. This decreased to 58% by three months and to 38% by six months (Table 4). Provision of commercial infant formula was pervasive throughout the six month follow-up period, while less than half of the participants received RUTF at discharge and only about a third at one and three months. The study monitored the type(s) of supplements provided but not the amount consumed.

Receiving any nutritional supplements at one or three months was not associated with a reduced or higher risk of persistent malnutrition.

## Access to services

Less than one-half of children ever received a WBOT home visit, and only 16% of participants were beneficiaries of the Zero Hunger programme during the six month follow-up period (Table 5).

Table 4. Provision of nutritional supplements based on discharge anthropometric status and type of supplement.

| Anthropometric category (at discharge) | Discharge | | One month | | Three months | | Six months | |
|---|---|---|---|---|---|---|---|---|
| | no. | % | no. | % | no. | % | no. | % |
| **No acute malnutrition** | (n = 14) | | (n = 14) | | (n = 14) | | (n = 13) | |
| Any | 14 | 100 | 9 | 64 | 6 | 43 | 5 | 38 |
| Commercial formula | 13 | 93 | 9 | 64 | 4 | 29 | 4 | 31 |
| Porridge | 2 | 14 | 3 | 21 | 5 | 36 | 5 | 38 |
| Ready-to-use therapeutic food | 5 | 36 | 2 | 14 | 2 | 14 | 0 | 0 |
| **Moderate acute malnutrition** | (n = 18) | | (n = 18) | | (n = 16) | | (n = 17) | |
| Any | 18 | 100 | 13 | 72 | 10 | 63 | 6 | 35 |
| Commercial formula | 18 | 100 | 11 | 61 | 7 | 44 | 6 | 35 |
| Porridge | 5 | 28 | 6 | 33 | 7 | 44 | 3 | 18 |
| Ready-to-use therapeutic food | 11 | 61 | 5 | 28 | 4 | 25 | 0 | 0 |
| **Severe acute malnutrition** | (n = 11) | | (n = 11) | | (n = 8) | | (n = 7) | |
| Any | 11 | 100 | 9 | 82 | 6 | 75 | 3 | 43 |
| Commercial formula | 11 | 100 | 8 | 73 | 4 | 50 | 2 | 29 |
| Porridge | 2 | 18 | 5 | 45 | 5 | 63 | 3 | 43 |
| Ready-to-use therapeutic food | 4 | 36 | 3 | 27 | 1 | 13 | 1 | 14 |
| **Any form of malnutrition (total)** | (n = 43) | | (n = 43) | | (n = 38) | | (n = 37) | |
| Any | 43 | 100 | 31 | 72 | 22 | 58 | 14 | 38 |
| Commercial formula | 41 | 95 | 28 | 65 | 15 | 39 | 12 | 32 |
| Porridge | 9 | 21 | 14 | 33 | 17 | 45 | 11 | 30 |
| Ready-to-use therapeutic food | 20 | 47 | 10 | 23 | 7 | 18 | 1 | 3 |

**Table 5. Access to support services at one, three and six months.**

|  | One month | Three months | Six months |
|---|---|---|---|
| **Zero Hunger programme** | 1/37 (3%) | 4/33 (12%) | 5/32 (16%) |
| **Ward based outreach team home visit (one or more)** | 11/42 (26%) | 16/37 (43%) | 13/36 (36%) |

## Discussion

This is the first report from an upper-middle-income country on the longer-term survival, recovery and relapse of children admitted and treated for SAM following hospital discharge. Most gratifying was the finding that none of the participants died. Rates of ongoing SAM and relapse rates were lower than reported in other low and middle income country settings [1, 6, 8, 13–15]. Nevertheless, almost half of the cohort remained malnourished (MAM or SAM) three months following discharge. Risk factors for adverse outcomes were similar to those identified in other low resourced settings, such as a low WLZ on admission or hospital discharge [1, 6, 14] and having an illness during the post-discharge recovery period [1, 8].

### Mortality

Studies reporting post-discharge mortality of children with SAM are limited and mainly emanate from low income settings. Mortality rates at 3–6 months of 2.7% to 8.7% have been described following discharge from outpatient and inpatient settings [3, 5, 7, 8]. A systematic review reported mortality rates between 0.6% and 10.4%, 6–24 months after discharge from community-based therapeutic feeding centres following nutritional rehabilitation for uncomplicated SAM [16]. To our best knowledge, only one study, from Pakistan [13], has previously reported a zero six-month mortality.

Various factors could have contributed to this positive outcome in our setting including (i) the presence of a district-based SAM intervention programme, (ii) the availability of nutritional supplements post-discharge (which has previously been shown to be protective against relapse and improve nutritional recovery) [1, 6, 7], (iii) accessible follow-up services at the clinic or hospital, and (iv) participant involvement in a study—more intensive follow-up with ready access to health care advice through mobile phones has proven benefit [17]. A recent meta-analysis indicated that biomedical (pancreatic enzyme supplementation and provision of symbiotics) and psychosocial interventions (child stimulation and parental education) help reduce SAM post-discharge mortality [18]. Although our study did not measure these, they may be beneficial to include in future research, because of the many underlying and multifaceted factors that may lead to mortality post discharge in SAM children.

### Relapse

Relapse data across studies are largely not comparable because of differences in discharge criteria, diverse post-discharge treatment protocols including supplementary food provision, variable follow-up periods, inconsistent definitions of relapse, and differing reporting of relapse as a point prevalence, cumulative prevalence, and incident rate [1, 2]. The study's relapse rate of six children (14%) with either ongoing or recurring SAM is on the lower end of reported incidence data. A 2019 systematic review indicated that 0–37% of children treated for SAM relapsed with most cases occurring within six months of discharge from treatment programme [1]. Factors contributing to the study sites zero mortality, as described earlier, were probably as relevant to this outcome.

## Morbidity and illness

The fact that more than half the children experienced an illness during the study follow up period is indicative of their enduring underlying immunosuppressed status despite recovering from SAM, as well as their ongoing exposure to social and environmental risk factors that may have led to their developing SAM originally. Participants illness profile (mostly fever, cough and diarrhoea) was similar to that reported in studies from Bangladesh [8], and Malawi [19], albeit with lower incidence rates. The latter study compared home-based RUTF to standard care in children with SAM, and found significantly fewer deaths, lower relapse rates and lower incidence of cough, diarrhoea among RUTF recipients at the two-month follow up [19]. The provision of nutritional supplements to our study participants may have assisted similarly.

A critically important study finding is that children who experienced an illness by three months following hospital discharge were significantly more likely to remain malnourished. This underscores the importance of offering specific disease prevention and health promotion advice to the caregivers of these children on discharge and at follow-up, even in better-resourced settings.

## Anthropometric recovery

Despite the laudable mortality and relapse rates, the anthropometric recovery of children with SAM was disappointing, with only just over half the children achieving a normal WLZ by three months. The critical regression period for growth was from one to three months, with almost half the participants demonstrating a reversal in mean WLZ (a -0.3 z-score reduction), and daily weight gain rate halving. During in-patient treatment for SAM a weight gain less than 5 g/kg/day is considered suboptimal [20]. No comparable standards exist for the post-discharge period, but the gains achieved by our cohort were clearly suboptimal.

The unsatisfactory anthropometric recovery outcome can mainly be attributed to a combination of inappropriate discharge criteria, concomitant new infections, inadequate or inappropriate nutritional supplementation and dysfunctional support services.

## Discharge criteria inappropriate

The current WHO recommendations are to transfer children from inpatient to outpatient care when their medical complications are resolved, they have a good appetite and are clinically well and alert. Furthermore children can be discharged from SAM treatment altogether when they are anthropometrically recovered (WLZ $\geq$-2 or MUAC $\geq$12.5 cm and no oedema for at least two weeks). The anthropometric indicator (MUAC or WLZ) that was used to diagnose SAM is used to evaluate whether the child has nutritionally recovered [2, 20].

Adhering to the WHO guidelines has been shown to be protective against relapse [1]. However, basing in-patient discharge decisions solely on clinical rather than anthropometric recovery criteria can have negative consequences. In our setting, this resulted in over two-thirds of children still being malnourished (MAM or SAM) at hospital discharge, with a higher risk of remaining malnourished at three months. This situation is pervasive in resource poor settings. In a recent study conducted in Zambia and Zimbabwe almost half of children had ongoing SAM at the time of discharge [6].

Our findings suggest that, in a setting where nutritional support for moderate malnutrition is provided after hospitalisation, children can be discharged from in-patient care once a WLZ of $\geq$-2.1, or a WLZ of >2.2 and MUAC of $\geq$12.0 cm, is achieved; if a persistent malnutrition (MAM or SAM) risk of 8% or 11%; respectively, at three months is considered acceptable.

Although longer hospital stays to allow for achievement of improved anthropometric recovery may be an obvious recommendation, this approach may itself have negative consequences

such as increased risk of nosocomial infection and longer hospital stay costs. A more cost-effective approach may be to provide better nutritional and social support following hospital discharge.

Interestingly, all eight children who presented with oedema at admission were anthropometrically normal by three months, confirming the different clinical and anthropometric course of this form of SAM. This finding has been described in other settings [4, 6].

## Nutritional supplements

A distinguishing feature of the study, compared to previous studies, was the high rates of nutrition supplement provision to children post-discharge with more than one-half still receiving it at three months. However, the sub-optimal improvements in anthropometric status raises concern about the effectiveness of the intervention. There was modest provision of RUTF compared to commercial formula. Greater use of RUTF rather than commercial formula can support better growth [21].

A 2020 systematic review indicated that the use of nutritional supplements with high-quality protein and adequate micronutrient content for three months improved anthropometric growth for children with MAM, with food products having more benefit than nutritional counselling and/or micronutrient supplementation [22]. This was particularly the case if supplementary food was of suitable quality and provided for an adequate period. Further research is warranted on what constitutes effective post-discharge supplementation. Measured outcomes should be functional extending beyond anthropometric proxies.

## Support services

A disappointing finding was the limited reach of the WBOT and Zero Hunger interventions. Their failure to engage in the district's SAM intervention project was recognised by the project team who lamented their inability to extract accountability [23]. All seven children who received a WBOT home visit by one month were still malnourished at three months suggesting that the WBOT's activities had minimal influence on growth outcomes. On a positive note, most caregivers (72%) were receiving a child support grant worth about $US30 (ZAR450)—a useful child survival and thriving intervention [1, 18, 24]–even though we were unable to demonstrate differences in persistent malnutrition rates in grant recipients. A single child who relapsed twice within six months was a non-South African national excluding him/her from benefiting from either a child support grant or the Zero Hunger intervention.

## Strengths and limitations

The study's greatest strength is its prospective nature and duration of follow-up. It is the first such among malnourished children in South Africa and also in any upper-middle-income country setting. Another strength is the high follow-up rate– 86% is particularly good for a setting such as where the study was done, reducing bias associated with 'lost to follow up which may undercount mortality and morbidity. The study's location in a district with an active SAM intervention project also allowed for assessment of components of that project. Few previous studies have explored anthropometric recovery in children with SAM supplemented for a three to six month period.

The study has several limitations. The greatest is the small sample size. This was the consequence of lower than predicted SAM hospitalisations during the dedicated study period. The zero mortality and low relapse rate prevented planned analysis of contributory risk factors for these outcomes. The primary researcher being the dietitian responsible for the care of some participants could have biased outcomes but this was considered unlikely, as no extraordinary

measures were instituted by her. We did not collect data on several potential risk factors for persistent malnutrition such as home food security, health seeking behaviour and adherence to nutritional supplementation. Neither did the study collect data on the children's feeding practices after discharge including breastfeeding and dietary diversity. Another potential limitation relates to seasonal variations in the risk of malnutrition. Although enrolment was spread over 10 months and included both winter (hunger) and summer (harvest) seasons, this could have influenced study outcomes.

The study findings cannot be easily generalised to other South African settings since few districts are implementing SAM projects similar to that undertaken in the study district. Nevertheless, project interventions such as nutritional supplements, WBOTs and social support programmes are ubiquitous in the country, although variably implemented.

## Conclusions and implications for policy and practice

Despite its limitations, the study highlights the potential of several interventions to improve post-discharge outcomes for children with SAM. Many children were still severely malnourished at discharge and this was associated with a poorer outcome. We have proposed a possible modification to the discharge criteria for adoption in the district. However, the response should not simply be to prolong the hospital stay to meet anthropometric discharge criteria. Instead, children could be risk stratified so that, for example, those with a lower admission WLZ or weaker home and community support structures are discharged later. Allowing children to be discharged to their closest facility may improve follow up rates and access to health care, thereby reducing relapse [1, 8, 14].

Post-discharge management is not just about food—this is necessary but not sufficient on its own. Further investigation of ideal nutritional supplementation programmes post discharge must be investigated. Importantly SAM programmes must work more closely with other child health services to form a more clearly defined and seamless functional continuum of integrated care. This is a tough ask and demands special effort to develop a collaborative work ethic, clear communication lines and common programme goals.

Identification of preventable risk factors for SAM relapse requires further research involving considerably larger sample sizes than the current study. Better understanding of the contextual factors that constrain the management of SAM at service points and at home are needed. Future research will also need to unravel exactly which post-discharge interventions are most effective and cost-effective for these vulnerable children, including exploring interventions addressing underlying risks for malnutrition such as household food security, caregiver well-being, and sub-optimal home environments.

## Acknowledgments

Our sincere thanks to all dietitians, doctors and nurses at Mamelodi and Bronkhorstspruit Hospitals and dietitians at Tshwane sub-districts 5 and 7 clinics for their invaluable support and contribution during patient enrolment and data collection. Dr Marshè Maharaj is recognised for offering supervision and guidance during data collection.

The study was undertaken in partial fulfilment of an MSc in Child Health at the University of the Witwatersrand, Johannesburg by AG.

## Author Contributions

**Conceptualization:** Angelika Grimbeek, Haroon Saloojee.

**Data curation:** Angelika Grimbeek, Haroon Saloojee.

**Formal analysis:** Angelika Grimbeek, Haroon Saloojee.

**Funding acquisition:** Angelika Grimbeek.

**Investigation:** Angelika Grimbeek.

**Methodology:** Angelika Grimbeek, Haroon Saloojee.

**Project administration:** Angelika Grimbeek, Haroon Saloojee.

**Resources:** Angelika Grimbeek, Haroon Saloojee.

**Software:** Angelika Grimbeek, Haroon Saloojee.

**Supervision:** Haroon Saloojee.

**Validation:** Angelika Grimbeek, Haroon Saloojee.

**Visualization:** Angelika Grimbeek, Haroon Saloojee.

**Writing – original draft:** Angelika Grimbeek, Haroon Saloojee.

**Writing – review & editing:** Angelika Grimbeek, Haroon Saloojee.

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
