## [Decision Letter · Decision Letter 0]

17 Nov 2021

PONE-D-21-31829Clinical and growth outcomes of severely malnourished children following hospital discharge in a South African settingPLOS ONE

Dear Dr. Peczak,

Thank you for submitting your manuscript to PLOS ONE. After careful consideration, we feel that it has merit but does not fully meet PLOS ONE’s publication criteria as it currently stands. Therefore, we invite you to submit a revised version of the manuscript that addresses the points raised during the review process, itemised below.

We look forward to receiving your revised manuscript.

Kind regards,

Brenda M. Morrow, PhD

Academic Editor

PLOS ONE

Journal Requirements:

Reviewers' comments:

Reviewer's Responses to Questions

**Comments to the Author**

1. Is the manuscript technically sound, and do the data support the conclusions?

Reviewer #1: Partly

Reviewer #2: Yes

2. Has the statistical analysis been performed appropriately and rigorously? 

Reviewer #1: Yes

Reviewer #2: Yes

3. Have the authors made all data underlying the findings in their manuscript fully available?

Reviewer #1: Yes

Reviewer #2: Yes

4. Is the manuscript presented in an intelligible fashion and written in standard English?

Reviewer #1: Yes

Reviewer #2: Yes

5. Review Comments to the Author

Reviewer #1: This prospective descriptive study gives a perspective on the follow up outcomes of children with SAM in on region in South Africa (referral hospital and one district hospital. There are limited recent data on this subject in southern Africa and South Africa.

The sampling strategy is well described with children fulfilling appropriate criteria. Omitting HFA <-3 as a criterion is appropriate as the causes and outcomes of stunting are largely not the same as those causing low WFL, MUAC and oedema.

The core of the paper relates to nutritional outcomes including the welcome outcome of no post-discharge mortality. The link of nutritional outcome to clinical status at discharge is a useful finding, given that discharge criteria in the two institutions did not coincide with WHO recommendation. I would recommend that these are recorded as planned secondary outcomes in the methods section. Only mortality is mentioned there, yet the bulk of the article is about other nutritional outcomes.

The persistence of SAM and significant nutritional deficits is an important finding, though muddied by the HIV and CP groups who did not do well. While I buy the idea that they should be included, reviewing the data without them might give more insight into standard patients with SAM who are the bulk of the cases in secondary and primary care situations which is where bang for buck in improved f/u strategies is to be found at population level.

While the paper gives extensive detail on discharge plans, nutritional inputs, follow up activities etc., including community follow up (rather dismal, it seems, and it does seem strange that with such small numbers involved one can discern that CHWs tended to worsen the situation. They may of course have only gone to those most at risk because the referring system made this more likely), because these are not standardised or grouped, their meaning in the context of the outcomes can't really be discerned for the large part - thus these detailed data are superfluous to the main purpose of the article (i.e. they don't provide guidance to others dealing with issues of SAM follow up - except to say that a more consistent strategy could be useful). Line 292 is an exception to this - though some statistical evidence of the non-association of nutritional supplements with nutritional outcomes would is required. This line is also misplaced - it should surely be under the Nutritional supplements heading (Line 299). [You say in the Conclusion that 'The study has provided evidence that continuing nutritional support through

supplementation may improve survival and reduce relapse rates in children with SAM following hospital discharge'. I suppose the key is word ''may' given line 291 which does not support statement this statistically (Type 2 error?). You are saying that the pervasiveness of supplementation is likely to be why there were not deaths and many children improved nutritionally, but the study does not prove this unfortunately..

Comments are made re stunting but HFA does not appear among the measures during f/u in the Methods.

The lack of association with variables listed towards the end of the Results section are likely to relate to small numbers as partial follow up data attrition started to bite (as you acknowledge in the Discussion).

Because this is a small sample (even though complete) in one region, I am recommending a revision to a short report, removing much of the dietary and other f/u detail which cannot be directly linked in a clear fashion to the nutritional outcomes (the main value of your work). (If you had been evaluating adherence to standard pre-discharge and f/u care of children with SAM, these details would have had more value.)

Reviewer #2: This paper describes post-discharge outcomes of children receiving inpatient treatment for severe acute malnutrition. This is an area of limited research, and hence this paper makes a useful contribution to the literature, particularly due to the 6 months of follow-up provided and unique setting of an UMIC of South Africa.

I recommend the paper be accepted after minor revision. This mainly relates to line 478 that the study 'has provided evidence that continuing nutritional support through supplementation may improve survival and reduce relapse rates in children with SAM following hospital discharge'. However it is not clear enough that the data supports this statement. There are many other results, such as the finding that better anthropometry at discharge is related to improved outcomes at follow-up, that could replace this.

There is good referencing and understanding of the literature throughout. The statistical analysis is appropriate for the data. An important addition to the literature on this topic.

6. PLOS authors have the option to publish the peer review history of their article (what does this mean?). If published, this will include your full peer review and any attached files.

Reviewer #1: No

Reviewer #2: **Yes: **Natasha O'Sullivan

---

## [Author Response · Author response to Decision Letter 0]

6 Dec 2021

Thank you for the comments from both reviewers. 

The details of these changes have been made and have been submitted as seen in the rebuttal letter and the manuscript with tract changes.

---

## [Decision Letter · Decision Letter 1]

4 Jan 2022

Clinical and growth outcomes of severely malnourished children following hospital discharge in a South African setting

PONE-D-21-31829R1

Dear Dr. Peczak,

We’re pleased to inform you that your manuscript has been judged scientifically suitable for publication and will be formally accepted for publication once it meets all outstanding technical requirements.

Kind regards,

Brenda M. Morrow, PhD

Academic Editor

PLOS ONE

Reviewers' comments:

Reviewer's Responses to Questions

**Comments to the Author**

1. If the authors have adequately addressed your comments raised in a previous round of review and you feel that this manuscript is now acceptable for publication, you may indicate that here to bypass the “Comments to the Author” section, enter your conflict of interest statement in the “Confidential to Editor” section, and submit your "Accept" recommendation.

Reviewer #1: All comments have been addressed

Reviewer #2: All comments have been addressed

2. Is the manuscript technically sound, and do the data support the conclusions?

Reviewer #1: Yes

Reviewer #2: Yes

3. Has the statistical analysis been performed appropriately and rigorously? 

Reviewer #1: Yes

Reviewer #2: Yes

4. Have the authors made all data underlying the findings in their manuscript fully available?

Reviewer #1: Yes

Reviewer #2: Yes

5. Is the manuscript presented in an intelligible fashion and written in standard English?

Reviewer #1: Yes

Reviewer #2: Yes

6. Review Comments to the Author

Reviewer #1: You have addressed the issues I raised in the review. I accept your contention that the additional factors that I felt were superfluous may be useful to readers and add some breadth to the literature on SAM outcomes. Especially as the journal does not have a Short Report option.

I read your discussion again and am happy that it covers the areas that I had recommended for omission clearly. On this reading there are a couple of punctuation errors that should be corrected:

Line 235 - remove comma

Line 419 - odd use of semi colons (they should be removed as they break the flow of the sentence). You might re-organise this sentence to increase readability - SAM and MAM are a long way from their associated figures.

Line 457 -add comma after 'excluding' to increase readability.

Reviewer #2: All reviewer comments sufficiently met and the paper is more concise and relevant, will be a very useful addition to the literature

7. PLOS authors have the option to publish the peer review history of their article (what does this mean?). If published, this will include your full peer review and any attached files.

Reviewer #1: No

Reviewer #2: **Yes: **Natasha O'Sullivan

---

## [Editor Report · Acceptance letter]

11 Jan 2022

PONE-D-21-31829R1 

Clinical and growth outcomes of severely malnourished children following hospital discharge in a South African setting 

Dear Dr. Grimbeek:

I'm pleased to inform you that your manuscript has been deemed suitable for publication in PLOS ONE. Congratulations! Your manuscript is now with our production department. 

Kind regards, 

on behalf of

Professor Brenda M. Morrow 

Academic Editor

PLOS ONE